# Regulation and Deregulation of Viral Gene Expression During High-Risk HPV Infection

**DOI:** 10.3390/v17070937

**Published:** 2025-06-30

**Authors:** Konstanze Schichl, John Doorbar

**Affiliations:** Department of Pathology, University of Cambridge, Cambridge CB2 1QP, UK; kas207@cam.ac.uk

**Keywords:** high-risk human papillomavirus, transformation zone, cervical crypt, microenvironment, transcriptional differences, cervical cancer, HPV gene expression

## Abstract

Cervical cancer remains a global health burden, with persistent infection by high-risk human papillomaviruses (HR-HPVs) being the primary etiological factor. HR-HPVs target stem-like cells of the cervical epithelium to establish chronic infections. Upon infection of the cervical transformation zone (TZ)—a region adjacent to the squamocolumnar junction (SCJ)—these viruses drive neoplastic transformation, which is due in part to the unique cellular composition and hormonal responsiveness of the TZ. Reserve cells, which can accumulate at the cervical crypt entrances of the TZ, are thought to be highly susceptible to HR-HPV infection because of their location beneath a single layer of columnar cells. Infection of the stratified ectocervical epithelium, in contrast, requires a wound to allow basal cell infection, replication, and the expression of early genes to adjust epithelial homeostasis while facilitating immune evasion. Persistent infection by HR-HPV types, particularly HPV16 and HPV18, can result in the deregulated expression of viral genes E6 and E7, driving cell cycle disruption, genomic instability, and subsequent viral genome integration. Differences in the microenvironment and transcriptional environment of the ectocervix compared with the TZ could explain the frequent deregulation of E6 and E7 at the latter site, which can drive disease progression towards cancer.

## 1. Introduction

Cervical cancer is one of the most common malignancies affecting women globally, with over 95% of cases linked to persistent infection by high-risk human papillomavirus (HR-HPV) types, including HPV16 and HPV18 [1]. These types have been classified as carcinogenic to humans (group 1) by the International Agency for Research on Cancer [2]. HR-HPVs depend on specific vulnerabilities in the cervical epithelium to establish chronic infections and can ultimately drive neoplastic transformation if not brought under control by the immune system. A critical site for HPV-associated carcinogenesis is the transformation zone (TZ) of the cervix, a region of dynamic epithelial remodelling where squamous and columnar epithelium converge adjacent to the squamocolumnar junction (SCJ) [3]. The TZ is particularly susceptible to HPV infection due to its unique histological structure and cellular composition, making it a hotspot for the development of cervical intraepithelial neoplasia (CIN) and, eventually, invasive cervical cancer [4,5].

The TZ is a region between the epithelium of the endo- and ectocervix that undergoes squamous metaplasia, a normal physiological process in which columnar epithelium is replaced by squamous epithelium under the influence of hormonal and environmental factors (Figure 1A). This remodelling process is prominent during puberty and early reproductive years, times of heightened HPV susceptibility [6]. The hormonal responsiveness of the TZ to oestrogen further exacerbates the vulnerability of this region to HPV-associated neoplasia. Oestrogen, which is involved in the epithelial homeostatic process of metaplasia, has been shown to modulate HR-HPV E6 and E7 expression, enhancing deregulated viral gene expression and promoting the progression of infected cells towards malignancy [7].

During oncogenesis, viral E6 and E7 act by targeting key tumour suppressor pathways, driving unchecked cell proliferation and genomic instability in the case of their deregulation. These viral proteins have a list of cellular targets that they use to modulate epithelial homeostasis and drive viral genome amplification. Some of these host cellular targets play a role in E6/E7 carcinogenesis when the viral proteins are deregulated. E6 modulates the expression of several host factors, including E6AP and tumour suppressor proteins p53 and NHERF1 [8,9]. Their downregulation modulates keratinocyte differentiation commitment and impairs apoptosis, promoting the survival of genetically unstable cells. In parallel, E7 inactivates the retinoblastoma protein (pRb) to initiate cell cycle entry [10], and degrades PTPN14 to limit keratinocyte differentiation [11]. While the modulation of epithelial homeostasis by the virus is essential for its life cycle, the deregulation of these processes can lead to the accumulation of genetic mutations, chromosomal aberrations, and ultimately malignant transformation.

Reserve cells are located under the columnar epithelium of the cervical TZ, often lining cervical crypt entrances (Figure 1A,B) [12], and have been proposed as multipotent progenitors capable of differentiating into squamous or columnar epithelial cells [13]. These undifferentiated cells can provide a permissive environment for HPV replication, and because of their similarity to epithelial basal calls, are thought to be the main target cell for the virus [14]. While reserve cells have similar characteristics to basal cells [12], it is believed that the difference in the microenvironment at the TZ compared to the stratified squamous epithelium means that the cells act differently upon interaction with the virus. Thus, the presence of reserve cells, coupled with active cell division during metaplasia, creates a vulnerable microenvironment for deregulated viral gene expression and oncogenic progression.

Therefore, the TZ of the cervix represents a uniquely susceptible site for HPV-mediated carcinogenesis. Persistent infections with HR-HPV types, particularly HPV16 and HPV18, amplify these vulnerabilities, driven by a deregulated expression of viral genes E6 and E7. In this review, we will elucidate the mechanisms of regulation and deregulation of the viral genes E6 and E7 upon HR-HPV infection of the cervical epithelium.

## 2. Control of Gene Expression During the Productive Life Cycle

### 2.1. Basal Cells as a Reservoir of Infection

HPV gene expression in basal epithelial cells is tightly regulated to facilitate viral episomal persistence, minimise immune detection, and maintain access to the proliferative compartment of the cervical epithelium to modulate tissue homeostasis in favour of the HPV life cycle. This regulation is achieved through tight promoter control, chromatin modifications, and the interaction with host cell factors (Table 1).

Basal cells play a crucial role in the productive life cycle of HPV, acting as the primary reservoir for viral persistence and replication [27]. At the stratified squamous epithelium of the ectocervix, HPV infects basal epithelial cells through microwounds in the mucosa or at the SCJ (Figure 2 [15]). It is estimated from tissue culture studies that the virus establishes its genome at an average of around 100 episomal copies per cell [21]. By establishing its episomal genome within the basal cells, HPV ensures long-term persistence. HPV16 and HPV31 strictly regulate episomal numbers by significantly suppressing STAT-1 with low levels of E6/E7 in infected undifferentiated cells [28]. Low expression of E1 and E2 has also been shown to play an essential role in maintaining viral DNA as an episome and in ensuring appropriate genome segregation during cell division [29]. The viral genome only expresses low levels of early genes to avoid immune detection, allowing the virus to maintain a reservoir in the basal layer. These low levels of gene expression may be considered comparable to the minimal ‘latency-associated’ transcript patterns characteristic of other viruses that cause chronic infections, and when coupled with late gene suppression, allow HPV to reach a state of clinical latency within the basal layer following clearance of a productive lesion [16]. However, our recent observations show a heterogeneous expression of E6/E7 in the basal layer, with some high-expressing cells and cells with expression below the detection threshold. While cells with high expression of E6/E7 are thought to retain or display accentuated stemness characteristics, low-expressing cells may be expected to leave the basal layer to progress through the viral life cycle [20].

Basal cells have a distinct transcriptional profile characterised by the presence of several endogenous transcription factors that play a critical role in regulating HPV gene expression and episomal maintenance. Among these, AP-1 (Activator Protein 1) and NF-κB (Nuclear Factor kappa-light-chain-enhancer of activated B cells) are key transcriptional activators of the HPV16 and 18 early promoters, which drive the expression of E6 and E7 [22,30]. AP-1, a heterodimer composed of Fos- and Jun-family proteins, binds to specific sequences in the LCR of the HPV genome to promote E6/E7 transcription. Similarly, NF-κB, a key regulator of immune and stress responses, also activates E6/E7 transcription via its binding to the LCR. However, in the presence of YY1—which is more highly expressed in the basal layer than the differentiated layers of the stratified epithelium—AP-1 and NF-κB activity are quenched. YY1 restricts E6/E7 expression by preventing AP-1 and NF-κB from binding the LCR through the formation of a CTCF (CCCTC-binding factor)–YY1-dependent chromatin loop, and the recruitment of polycomb repressor complexes (PRC1 and PCR2) [31,32,33].

p63, a transcription factor of the p53 family present in undifferentiated basal cells of the epithelium, is a well-established marker of epithelial stratification potential. ΔNp63, the predominant isoform in basal keratinocytes, is essential for keratinocyte proliferation, stemness, and the suppression of inflammation (as reviewed in [34]). HPV exploits p63 to maintain its genome as a stable episome, which p63 indirectly supports by promoting cell survival, inhibiting differentiation, and maintaining a replication-competent environment in basal cells. HPV E7 stabilises ΔNp63 expression by interfering with pRb, which normally suppresses p63 activity, leading to the release of E2F transcription factor to induce expression of S-phase genes, extending the lifespan of the basal cells by preventing their differentiation and favouring proliferation [19,35]. Additionally, p63 regulates the expression of cellular factors necessary for viral genome tethering, such as Brd4 (Bromodomain-containing protein 4), which interacts with HPV E2 to anchor viral episomes to host chromatin to ensure equal partitioning of the genome to both daughter cells [36,37], and is important to prevent the proteasomal degradation of E2 [38]. This prevents the excessive loss of HPV genomes during basal cell division, allowing the virus to persist.

The viral E2 protein is also a key regulator of early promoter activity and acts as a repressor of E6 and E7 expression by binding E2-binding sites (E2BSs). In HPV16, E2 binds to four specific E2BSs (E2BSs 1–4) located upstream of the early promoter in the upstream regulatory region (URR) [39]. This physically blocks transcription factor access, reducing the activity of the early promoter. As a result, low levels of E6 and E7 are maintained, allowing viral persistence in the basal cells without triggering host immune defences. On the other hand, E2 can form a complex with E1, which is a DNA helicase that recruits key cellular replication proteins when associated with E2 at the E2BSs [40]. 

HPV16 and 31 replication is restricted by the highly conserved E8^E2 repressor protein, which is a fusion between E8 and the C-terminal half of E2. E8^E2 restricts replication by interacting with the NCoR/SMRT (Nuclear Receptor Corepressor/Silencing Mediator of Retinoid and Thyroid Hormone Receptors) corepressor complex via the E8 region and at least one E2BS via the E2 region to limit viral replication in the undifferentiated cells [41,42]. The NCoR/SMRT corepressor complex represses gene transcription by recruiting histone deacetylases—importantly HDAC3—and other chromatin-remodelling factors, tightening chromatin structures and preventing RNA polymerases and transcription factors from binding. E8^E2-mediated transcriptional inhibition of the early promoter was shown to be more efficient than that of E2 alone [43]. Additionally, HPV16 E8^E2 not only represses viral replication in the undifferentiated layers but also restricts productive replication in differentiated epithelia [42]. These findings suggest that a balanced expression ratio of E8^E2 to E2 is important in the regulated expression of E6 and E7 in basal and mid-epithelial layers.

HPV16 promotes the persistence of infected keratinocytes in the basal and parabasal layers by downregulating Notch expression through its ability to drive E6AP-mediated degradation of p53—a transcriptional activator of Notch—which can induce keratinocyte commitment to differentiation [24,44]. Similarly, Mus musculus papillomavirus (MmuPV) studies using cell competition assays showed an important role of E6 in facilitating the persistence of infected cells in the epithelial basal layer through its interaction with MAML1, another Notch transcriptional activator [45]. Through its interference with the Notch pathway, HPV16 E6 and E7 independently modulate the homeostasis of the infected cells [25,46].

Strict regulation of viral gene expression in basal cells is an evolved function that is important for a successful viral life cycle. Initially, it prevents premature viral replication in undifferentiated cells to optimise virion production. Additionally, low-level early gene expression helps the virus avoid immune detection, aiding its persistence in the basal layer as a result of epithelial homeostasis modulation. Finally, the tight regulation of gene expression prevents genomic instability and ensures episomal maintenance rather than viral integration. This well-orchestrated regulatory mechanism allows HPV to persist in basal cells while minimising the risk of early immune clearance or oncogenic transformation during persistent infection.

### 2.2. Genome Amplification and Virus Production

HPV gene expression is tightly linked to the differentiation state of the host epithelial cell. E6 and E7 are expressed in the basal cells at low levels to modulate host cell cycle regulation, ensuring the continued proliferation of infected basal cells, and creating a supportive environment for episomal maintenance and replication. As basal cells divide, HPV genomes are partitioned between the daughter cells, some of which remain in the basal layer, while others differentiate and migrate toward the epithelial surface [47]. In stratified epithelia, it has been suggested that the high-density environment upon cell division triggers the basal cells to leave the basal layer by delamination [48]. This process can be modulated by HPV E6 through its suppression of the p53 function [20]. The viral genome is amplified at low levels during each S-phase of the basal cells and returns to pre-S-phase levels during mitosis, regulated by tethering of the viral genome to the host chromosome, and loss of viral genomes to the cytosol [49].

In undifferentiated cells, such as basal and reserve cells, transcription is primarily driven by the early promoter(s) (p97 in HPV16; p105 in HPV18), resulting in the low-level expression of early genes (E1, E2, E4, E5, E6, and E7). During this phase of the viral life cycle, late gene expression is prevented—among other mechanisms—by the termination of the transcripts at early polyadenylation sites (pAEs) located downstream of the E5 gene [50]. As the infected epithelial cells differentiate, a shift from early to late gene expression occurs. This involves a switch in promoter activity, the late promoters (p670 in HPV16; p811 in HPV18) become activated, and a shift in polyadenylation site usage occurs [51]. In differentiated cells, pAEs are bypassed through alternative splicing, producing late transcripts, which do not include the early pAEs. This shift is facilitated by the downregulation of polyadenylation factors that enhance usage of the early pAEs, and by the reduced expression of YY1—a suppressor of the late promoters [50,52]. As a result, viral capsid proteins L1 and L2 are strongly expressed only in the upper, differentiated layers of the epithelium. This tightly controlled switch ensures that virion production is confined to the upper epithelial layers, minimising immune detection.

To maintain a replication-competent environment in the differentiated keratinocytes, HR-HPV express E7 in the mid-epithelial layers, inactivating pRB and the related proteins p107 and p130 that control S-phase re-entry following cell cycle exit [18,53] to allow viral genome amplification. The late promoter activity is increased in suprabasal layers, leading to the expression of structural proteins L1 and L2, which are required for the assembly of new viral particles, as well as E4, which reorganises the keratin network to favour virion assembly [26]. This differentiation-dependent regulation ensures that virion production only occurs in the upper layers of the epithelium, where immune surveillance is generally reduced.

HR-HPV modulate keratinocyte differentiation in the mid- to upper-epithelial layers, allowing it to amplify viral genomes, prevent cell cycle exit, and prepare virions for shedding. By blocking Notch, p53, and pRb, the virus ensures that infected cells remain in the S-phase, supporting viral replication while still migrating upwards. The tight regulation of differentiation and proliferation of the epithelial cells is essential for persistent infection and high viral yield.

## 3. Deregulation of Viral Gene Expression

### 3.1. Deregulation Occurs at Specific Epithelial Sites—Hotspots

CIN arises predominantly in specific regions of the cervix where the epithelial environment is particularly susceptible to HPV infection and subsequent deregulation of viral gene expression (Figure 1C). These hotspots are defined by their unique cellular architecture that provides a portal of entry for the virus, hormonal and immune responsiveness, and vulnerability to persistent HPV infection.

The TZ is a critical site for HPV-associated neoplasia and is where most CIN lesions develop due to its unique cellular organisation and epithelial architecture [5]. The reserve cells are a cell type uniquely present at the TZ, defined by their subcolumnar location and unique biomarker expression pattern (cytokeratin 5, cytokeratin 17, and p63) [12]. They share some characteristics with the basal cells of the squamous epithelium, such as their ability to differentiate and divide; however, in their quiescent state, they have no ability to become stratified due to a local Wnt gradient [54]. Their location under a singular layer of columnar cells might make them an easier target for HPV infection than basal cells as their infection does not require the initial breaking of the epithelial barrier (Figure 2A). The SCJ at the TZ has also been identified as a vulnerable site to HPV infection [3]. It is a small area forming the active interface between columnar and squamous epithelia. The cells at the SCJ of the TZ undergo squamous metaplasia, where the columnar epithelium is replaced by immature squamous epithelium. These SC junctional cells may give rise to reserve cells through transdifferentiation [55], implying a similar vulnerability to infection by HR-HPV types. However, Regauer and Reich [4] suggest that reserve cells arise during foetal development, and are derived from Müllerian duct epithelium, which suggests that the SC junctional cells are distinct from reserve cells in their origins. Recent findings also suggest the expression of cytokeratin 7 (CK7) as a marker of a subset of cells within the TZ, particularly at the SCJ, which may be especially prone to HPV-mediated neoplasia. These cells have been reported to share features with embryonic progenitor cells and are often found in early cervical lesions, supporting the hypothesis that they represent a transformation-prone cell population with high susceptibility to oncogenic HPV infection [56]. However, our work did not find that CK7 alone marks a specific subset of SC junctional cells [12].

Cervical crypts, which are an important feature of the TZ, appear from our studies to be hotspots for the development of HPV-related neoplasia (Figure 3). It is especially evident that HPV16-infected cells at the TZ become more dysplastic during disease progression when compared to infections at other epithelial sites, and infections with other HR-HPV types [57]. Cervical crypts are important structural elements of the cervical epithelium, which are present at the TZ and parts of the endocervix. The cervical crypts are gland-like clefts lined with mucin secretory columnar cells that run across the transformation zone longitudinally and transversely [58]. While the developmental origins of cervical crypts are poorly understood, we can infer from other epithelial sites with similar morphogenesis that they might involve transcriptional differences that will also be apparent at the TZ. In the mammary gland, FOXA1 plays an essential role in ductal morphogenesis by regulating the expression of oestrogen receptor alpha (ERα) [59]. Similarly, FOXA1 is integral to the development of eccrine sweat glands by regulating genes involved in sweat secretion, such as those encoding ion channels and transporters [60]. As the TZ is considered one of the most vulnerable epithelial sites to develop HR-HPV-induced neoplasia [5], elucidating those differences in the microenvironment at the TZ and more specifically the crypts is important for understanding the viral pathology. The reserve cells, which are found at the entrances of crypts, are believed to be the main target cell for HPV infections in the transformation zone due to their similarities to basal cells [14]. However, Stanley [61] hypothesised that the reserve cells may be only semi-permissive to the virus, not supporting the late stages of the productive life cycle. Thus, infection of reserve cells by HR-HPV would cause abortive infections, often at the entrances of cervical crypts where they can accumulate. In the oropharynx, HPV preferentially infects tissue stem cells at the tonsillar crypts (Figure 2B), where virus-specific T cell activity is inhibited by the virus to facilitate immune evasion during initial infection [62]. The cells lining the tonsillar crypts express PD-L1 (Programmed Death Ligand-1) at their surface, while the majority of site-infiltrating CD8+ T cells express high levels of PD-1 (Programmed Cell Death-1), which suppresses the excessive activation of T cells and may result in an immune-privileged site for viral infection and tumourigenesis [62]. While tonsils are functionally quite different from the cervix, which does not have a reticulated epithelial crypt structure, it has been suggested that the cervical TZ might have a different immune response to viral infections than other sites in the epithelium. Thus, similar immunological control mechanisms might exist at the cervical crypts of the TZ, turning them into immune-privileged sites—hotspots for HPV infection and the subsequent deregulation of E6 and E7 expression. For example, it has been suggested that the density of Langerhans cells is decreased in the TZ compared to the ectocervix [63]. Meanwhile, the single-layer structure of the simple columnar epithelium at the endocervix (and the reticulated epithelium of the tonsillar crypts (Figure 2B)) may facilitate viral entry into conveniently located reserve cells, while the confined architecture of the cervical crypts may facilitate viral persistence and immune evasion.

In summary, hotspots for deregulated viral gene expression and CIN formation include the transformation zone, the squamocolumnar junction and newly formed metaplastic epithelium, and the cervical crypt entrances. These regions share common features of hormonal responsiveness, and possibly low immune surveillance, providing an optimal environment for HPV persistence and oncogenesis.

### 3.2. Deregulation and Why It Happens

The deregulation of HR-HPV E6 and E7 expression is critical for the progression from infection to neoplasia. Transcriptional control of E6 and E7 is orchestrated by a combination of host factors, including viral regulatory proteins, host transcription factors, epigenetic modifications, and the differentiation state of the host epithelial cells. Disruptions of these regulatory mechanisms contribute to a deregulation in the expression of E6 and E7, contributing to their oncogenic properties and favouring carcinogenesis [64].

In a productive infection, the expression of E6 and E7 is meticulously regulated and primarily localised to the basal layer to regulate homeostasis and genome maintenance, and the mid-epithelial layers are regulated to control viral genome amplification. Deregulation involves the overexpression, and uncontrolled, persistent expression of E6 and E7 across all epithelial layers. This disrupts key cellular pathways and leads to the non-completion of the viral life cycle, giving a potent competitive advantage to the infected cells when compared to their neighbours [65,66]. As a result, the virus undergoes an abortive life cycle, which does not produce infectious viral particles, and the fast-dividing cells accumulate mutations, eventually leading to cancer.

A well-established mechanism of deregulation involves the loss of E2-mediated transcriptional repression of E6 and E7, shifting transcription towards their overexpression. Normally, the HPV E2 protein inhibits the activity of the early promoter responsible for E6 and E7 transcription by binding to specific E2BSs in the LCR. However, in many cervical carcinoma cases, the viral genome is integrated into the host genome, disrupting the E2 open reading frame, leading to a constitutive expression of E6/E7 [64]. Additionally, epigenetic modifications of E2BSs, such as hypermethylation, can mimic the loss of repressive function of E2 even in episomal HPV infections, leading to deregulated E6/E7 expression. CpG methylation status of the HPV URR has been directly linked to changes in p97 activity, the promoter regulating E6 and E7 expression. CpG hypermethylation in the HPV16 E2BSs of the URR has been shown to abolish E2-binding [67], especially at E2BS2 [68], which is integral for the regulation of p97 activity. Studies in W12 cells with episomal HPV16 showed that methylation of the viral LCR decreases with cellular differentiation [68]. Chromatin immunoprecipitation (ChIP) assays have shown that open chromatin configurations, characterised by H3K4me3 and H3K27ac at the HPV early promoter, correlate with high E6/E7 expression levels [23].

Additionally, host transcription factors play an important role in modulating E6 and E7 expression. The activation of the AP-1 complex (modulated by the signals from the EGFR/MEK/ERK cascade) and the resulting binding of AP-1 family members—such as Jun and Fos—to the LCR enhance early gene transcription [22]. Other transcription factors such as NF-κB and HIF-1α have been shown to contribute to the upregulation of E6 and E7 under conditions of oxidative stress and chronic inflammation [69].

Amongst YY1, NF-κB, and other previously mentioned transcription factors, FOXA1 was suggested to have an important role in the modulation of HPV gene expression and the subsequent driving of HPV-induced cervical neoplasia [70]. FOXA1 directly binds the long control region (LCR) of several HR-HPV types, including HPV16 and HPV18. By binding the LCR, FOXA1 activates the early promoter, significantly enhancing the transcription of E6 and E7 [70]. Moreover, FOXA1 is differentially expressed in different stages of the disease, suggested by an increase in its expression in immortalised and HPV-transformed cell lines compared to normal cells [71], as well as an increased expression in cervical carcinomas [72], implying an important role in cervical disease progression. FOXA1 expression is strongly linked to oestrogen, mainly through its interaction with oestrogen receptors (ERs). In breast cancer, FOXA1 was found to facilitate ERα binding to chromatin, thereby modulating oestrogen-responsive gene expression [73]. In the cervix, ERs are weakly expressed in the endocervix but strongly expressed in the ectocervix and cervical crypts [6,74], and ERα was identified as one of the main forms present [75]. HPV E6 and E7 expression is also heavily influenced by oestrogen, often correlated with the upregulation of the oncogenes and, therefore, their deregulation. The HPV16 and HPV18 genomes contain sequences resembling oestrogen response elements (EREs) in the LCR, allowing the oestrogen–receptor complex to bind and upregulate viral gene expression [7], a mechanism which might be aided by FOXA1. Our recent findings suggest that FOXA1 is more highly expressed in the cervical crypt in the presence of HPV than on the surface epithelium, emphasising the importance of the cervical crypt for promoting the deregulation of E6 and E7 expression, as well as neoplastic development. Collectively, these findings suggest that the interplay of FOXA1 with oestrogen might be important in the deregulation of HR-HPV E6 and E7 expression, especially at the cervical crypts, where both ERs and FOXA1 are more prevalent than other sites of the cervical epithelium. This evidence suggests that there might be a transcriptional environment characteristic of the crypt entrance.

The infection of semi-permissive cells—cells that are permissive for early gene expression and viral DNA replication but not late gene expression [61]—can also contribute to the deregulation of viral gene expression. For an HPV infection to produce viral particles, the virus must infect a cell with differentiation potential as viral replication is linked to the differentiation state of the epithelial cells [15]. However, when HPV infects semi-permissive cells, such as columnar cells, this control mechanism becomes disrupted. In semi-permissive environments, the viral genome may persist as an episome with low-level transcriptional activity, or even undergo integration events, leading to deregulated viral gene expression [76]. This is particularly relevant for HPV18, which shows a greater tendency to persist and integrate in semi-permissive cells and a strong tropism for glandular epithelium, the cellular origin of adenocarcinoma in situ [77]. The loss of differentiation-dependent repression of E6 and E7 can lead to their constitutive expression, promoting cell cycle re-entry and genomic instability, leading to HPV-driven carcinogenesis [78].

Collectively, deregulated E6 and E7 expression orchestrates a long list of molecular events that drive malignant transformation. By subverting cell cycle checkpoints, promoting genomic instability, and dampening host immune responses, HPV-infected cells acquire oncogenic potential, paving the way for neoplastic progression and eventual tumour development [79]. Further elucidation of these regulatory networks may, amongst other things, yield novel therapeutic targets for HPV-associated malignancies.

### 3.3. Consequences of Deregulation

The deregulation of E6 and E7 expression leads to a variety of cellular changes causing genomic instability and cell cycle deregulation in host cells that set the stage for malignant transformation [80]. E7 induces centrosome duplication errors, which promote chromosomal instability and aneuploidy, a hallmark of cancer, while E6 promotes the accumulation of supernumerary centrosomes, causing additional chromosomal instability [79,81]. Additionally, increased E7 expression resulting from HPV16 integration has been linked to chromosomal instability in host cells [82]. HPV16 E6 upregulates telomerase reverse transcriptase (hTERT), impairing telomere maintenance and extending the life span of infected keratinocytes independently of the E6-mediated degradation of p53, making it an important component of the HPV-induced transformation [83]. At an epigenetic level, HR-HPV E6 and E7 alter host microRNA (miRNA) profiles [84]. Harden and colleagues [85] identified the deregulation of 51 miRNAs and 1456 potential target mRNAs with the expression of HPV16 E6 and E7, which were involved in pathways related to immune evasion, epithelial–mesenchymal transition (EMT), and apoptosis resistance [86].

Integration of the HPV genome into the host’s DNA is not a natural part of the viral life cycle but rather a pathological event that occurs in some persistent infections [87]. This process is typically triggered by DNA damage resulting from the constitutive, deregulated expression of E6 and E7, which creates breaks in both viral and host DNA. The instability of the episomal form of the viral genome promotes its integration as a compensatory mechanism. Notably, HPV18 integrates into the host genome both earlier during the progression and more frequently than other HR-HPV types [88]. While integration is observed in approximately 13–20% of HPV16-positive lesions, it occurs in as many as 59–80% of HPV18-positive lesions graded from CIN1 to cancer [89]. Once integrated, HPV loses the ability to complete its normal productive cycle as late genes (L1, L2, and E4), necessary for capsid protein production and the assembly of infectious virions, are no longer expressed. The infected cells continue to proliferate uncontrollably without differentiating, leading to the formation of CIN and eventually invasive cervical cancer.

Additionally, the deregulation of viral gene expression impacts the immune microenvironment. Constitutive E6 and E7 expression modifies immune signalling pathways, leading to immune evasion, and contributes to the incidence of HPV-associated neoplasia. HPV16 E6 was shown to downregulate type I interferon responses and reduce the expression of antigen-presenting molecules including MHC class I, allowing infected cells to escape immune surveillance [90]. HPV16 E7 evades the immune system by suppressing interferon signalling and reducing antigen presentation. In a non-inflammatory epithelial context, it creates a tolerogenic environment that inhibits dendritic cell maturation and T-cell activation. Additionally, E7 downregulates pro-inflammatory cytokines and interferes with innate antiviral signalling [90,91]. HPV E5 also plays a critical role in immune evasion by downregulating the surface expression of MHC class I and II molecules, impairing antigen presentation to both CD8+ and CD4+ T cells [92]. E5 interferes with endosomal acidification and impairs the trafficking of immune molecules, further reducing immune recognition of infected cells [93].

In addition to immune evasion mechanisms, viral E6 and E7 can induce changes in the tumour microenvironment, further driving neoplastic development and progression to cancer. The deregulated viral gene expression promotes EMT—a process characterised by the loss of epithelial markers and the gain of mesenchymal markers—which may eventually lead to tumour invasion and metastasis [94]. Deregulation of E6 and E7 has also been linked to the upregulation of angiogenic factors, such as VEGF, additionally supporting metastasis and tumour growth [95,96].

Thus, the deregulation of viral gene expression disrupts the balance between viral persistence and productive replication, disrupting epithelial homeostasis and promoting oncogenesis. Viral genome integration amplifies these effects by stabilising E6 and E7 overexpression, ultimately driving the transition from HPV infection to malignancy. However, the relatively high rates of HPV-positive cervical cancers containing only episomal viral DNA [97] suggest the importance of the other mechanisms linking the deregulation of viral gene expression to malignant disease progression.

## 4. Summary

HR-HPVs can infect at the cervical TZ, a highly susceptible region where the columnar epithelium of the endocervix meets the stratified squamous epithelium of the ectocervix that lies adjacent to the SCJ. The TZ is a dynamic region, which can undergo epithelial remodelling via cellular metaplasia, and it has been identified as one of the most vulnerable sites for the development of HPV-induced neoplasias.

Within the TZ, reserve cells are the main target for HPV, which provide a permissive environment for viral persistence due to their basal-like characteristics. Upon infection, HPV establishes its genome as an episome, replicating and expressing early genes at low levels to evade immune detection. However, due to microenvironmental differences in the TZ location of the reserve cells compared with the ectocervical location of the basal cells, their persistent infection often results in the deregulated expression of viral E6 and E7 genes, facilitating possible genome integration and neoplastic transformation.

Deregulated expression gives E6 and E7 their oncogenic properties, while it is still poorly understood how this deregulation occurs. The deregulated expression of E6/E7 drives genomic instability and disrupts epithelial homeostasis, ultimately driving malignant transformation. The hormonal responsiveness of the TZ to oestrogen further exacerbates the deregulation of viral gene expression at the site.

A critical anatomical feature within the TZ is the cervical crypt, which has emerged as a hotspot for HR-HPV infection and deregulated gene expression. The cervical crypts provide structural niches where reserve cells are more highly concentrated and are more easily exposed to infection due to the single-layered columnar epithelium overlaying them. The identified transcriptional differences and potential differences in the immune environment may facilitate persistent viral infection and promote deregulated viral gene expression.

To summarise, a combination of permissive cellular niche (reserve cells), microenvironmental differences (hormonal responsiveness and transcriptional differences), and potential immune evasion mechanisms create an environment at the TZ that is prone to persistent HR-HPV infection and deregulated viral gene expression. These insights emphasise the importance of site-specific differences in disease progression and outcomes.

## Figures and Tables

**Figure 1 viruses-17-00937-f001:**
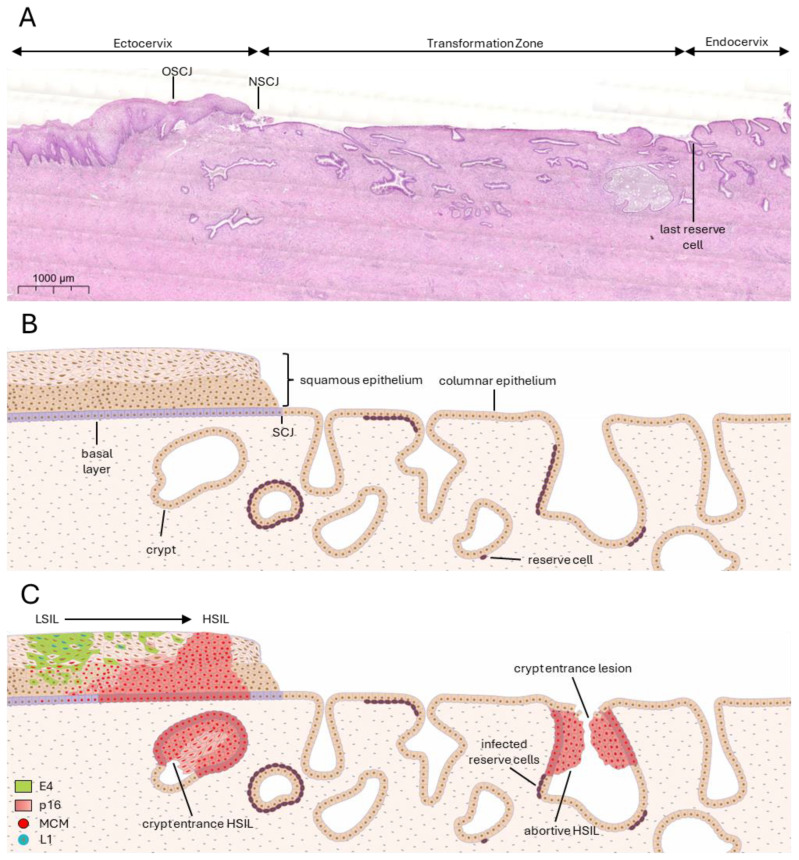
Human papillomavirus cervical infection sites and lesion formation. (**A**) Haematoxylin and eosin stain of an HPV-negative cervical biopsy sample spanning from ectocervix to transformation zone to endocervix. The original squamocolumnar junction (OSCJ) is found within the ectocervix, before the first cervical crypt. Through hormonal influences leading to metaplasia events, the SCJ migrates, forming the new SCJ (NSCJ). The transformation zone begins at the NSCJ to the last reserve cell, where the endocervix begins. (**B**) Illustration of ectocervix and transformation zone of an uninfected cervical epithelium. (**C**) Illustration of ectocervix and transformation zone of an HPV-infected cervical epithelium. Low-grade squamous intraepithelial lesions (LSILs) are most often found at the stratified epithelium of the ectocervix, while high-grade SILs (HSILs) are also found in cervical crypt entrances. MCM and p16 are surrogate markers for high-risk HPV infection and, in abundance, indicate deregulation of E6/E7 expression. Presence of E4 and L1 indicates productivity of the infection.

**Figure 2 viruses-17-00937-f002:**
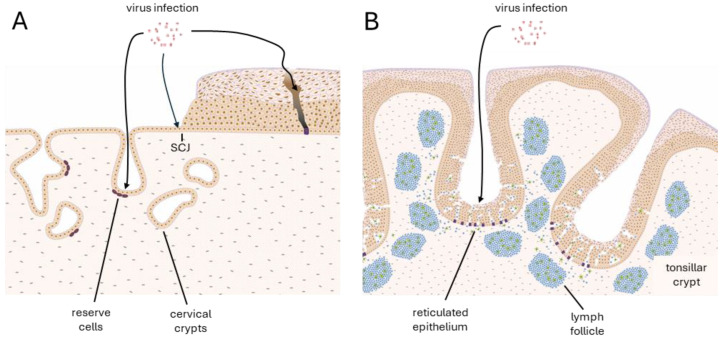
Human papillomavirus routes of entry. (**A**) Human papillomaviruses require a wound to access the basal layer of stratified epithelial tissue such as the ectocervix. Infection may be facilitated at the cervical squamocolumnar junction (SCJ) and at columnar epithelial sites that overlay cells such as the cervical reserve cells, which are HPV target cells. Reserve cells are usually found lining cervical crypts. (**B**) In the oropharynx, infection and neoplasia generally occur at the reticulated epithelium of the tonsillar crypts.

**Figure 3 viruses-17-00937-f003:**
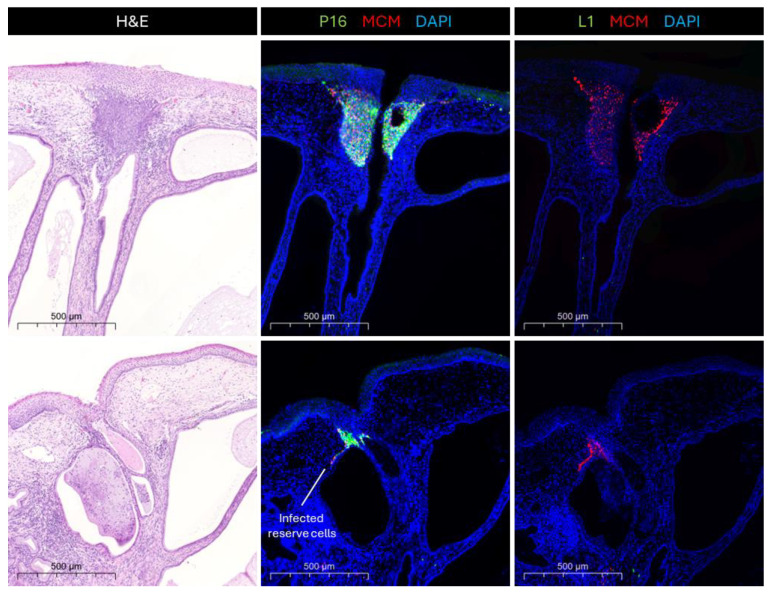
Cervical crypts—hotspots for neoplasia. Cervical crypt entrances harbour HSILs, which are identified by the expression of infection surrogate markers MCM and p16 at the full thickness of the lesion. MCM and p16 are markers for cell cycle entry, whose expression correlates with the expression of HPV E6 and E7. Additionally, these lesions are usually abortive, which is indicated by the absence of the viral capsid protein L1. A single layer of infected reserve cells tailing off from the neoplastic lesion can sometimes be seen.

**Table 1 viruses-17-00937-t001:** Regulatory mechanisms of a successful HPV life cycle. Summary of key regulatory mechanisms of a productive viral infection and their desired outcomes for the HPV life cycle.

Epithelial Layer	Key Regulatory Mechanisms	Outcomes
**Basal Layer** (stem-like cells, site of HPV infection)	**Transcriptional Control:** Early gene expression (E6/E7) is kept low to prevent immune activation [15,16]. E1 and E2 initiate episomal maintenance and modulate genome partitioning [15]. **Epigenetic Regulation:** DNA is tightly packed with histones to minimise viral gene expression [17]. **Epithelial Homeostasis:** E6 and E7 increase longevity of infected cells [18,19]. E6 degrades p53 via E6AP, inhibiting cell delamination [20]. E7 modulates cell cycle progression [15]. **Viral Genome Maintenance:** The viral genome is maintained as low copy number episome [15,21].	HPV establishes reservoir that is not detected by the immune system. Viral genomes are copied and passed to daughter cells.
**Parabasal Layers** (early differentiating cells)	**Transcriptional Control:** Early gene expression can decline. Cell division declines [22]. **Epigenetic Regulation:** DNA methylation patterns change to allow viral replication [17,23]. **Cell Signalling Modulation:** HPV alters Notch and Wnt pathways, delaying terminal differentiation [24,25].	Infected cells continue to undergo cell division, ensuring viral genome replication.
**Mid-epithelial Layers** (late differentiating cells)	**Early Gene Expression:** Increased E7 expression to maintain replication-competent environment [18]. E5 enhances amplification via EGFR signalling and MAP kinase activation [15]. **Late Gene Activation:** Expression of viral structural proteins (L1, L2) through alternative splicing, bypassing early polyadenylation sites [15]. **Cytoskeletal Remodelling:** E4 reorganises keratin network to favour virion assembly [26].	High levels of viral genome amplification. Initiation of late gene expression.
**Upper-epithelial Layers** (fully differentiated keratinocytes)	**Virion Assembly:** L2 is recruited to replication foci by E2, then L1 and L2 encapsidate viral genomes [15]. **Virus Maturation:** Superficial keratinocytes undergo redox state change to facilitate capsid maturation. Disulphide bonding between L1 proteins [15]. **Immune Evasion:** Terminally differentiated keratinocytes lack immune surveillance, allowing mostly undetected viral shedding [15,27].	HPV virions are assembled and shed with keratinocytes. The infection remains undetected by the immune system.

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
