# Peer review of "Regulation and Deregulation of Viral Gene Expression During High-Risk HPV Infection"

_viruses, 2025, doi:10.3390/v17070937_

Round 1
Reviewer 1 Report
Comments and Suggestions for Authors
This is a very well written and thoughtful review about HPV infection of the cervix. I have no specific edits or comments except perhaps to change the title to better describe the content of the manuscript.
Reviewer 2 Report
Comments and Suggestions for Authors
I thank for the opportunity to review this comprehensive and well written review on HPV viral gene expression.
My only comment is the suggestion to include some data on HPV18, due to the propensity of this type to cause glandular lesions and to integrate early into the cellular genome. These characteristics make its natural history somehow different from that of HPV16 (and the other high-risk types), and less efficient the prevention of HPV18-associated lesions by screening.
Reviewer 3 Report
Comments and Suggestions for Authors
The review by Konstanze Schichl and John Doorbar is very interesting because it summarizes the current knowledge on the regulation and deregulation of viral genes during HPV infection and tries to explain the transition from a productive infection to a persistent one until the cellular transformation following the deregulation of viral expression. The review is well written but, in my opinion, before its publication, it needs the authors to introduce paragraphs that discuss these two points:
1.The expression of CK7 seems to be a marker of cells (especially in the TZ) in which HPV infection can have an evolution to transformation and cancer (Mod Pathol 29, 1501–1510 (2016). https://doi.org/10.1038/modpathol.2016.141). This hypothesis should be included and commented on.
- Immune evasion is a key point of the infection and the oncogene E5 (pivotal works by Saveria Campo: Mol Cancer. 2011 Nov 11;10:140. doi: 10.1186/1476-4598-10-140.) seems to play an important role (the authors only mention the activity of E7). The function of E5 in immune evasion should be added and commented on.
Round 2
Reviewer 3 Report
Comments and Suggestions for Authors
Paper was improved and it is suitable for publication